# A distal regulatory region of a class I human histone deacetylase

Nicolas D. Werbeck[1,3,4], Vaibhav Kumar Shukla [1,4], Micha B. A. Kunze [1], Havva Yalinca[1], Ruth B. Pritchard [1], Lucas Siemons [1], Somnath Mondal[1], Simon O. R. Greenwood [1,2], John Kirkpatrick [1], Charles M. Marson[2] & D. Flemming Hansen [1✉]

Histone deacetylases (HDACs) are key enzymes in epigenetics and important drug targets in cancer biology. Whilst it has been established that HDACs regulate many cellular processes, far less is known about the regulation of these enzymes themselves. Here, we show that HDAC8 is allosterically regulated by shifts in populations between exchanging states. An inactive state is identified, which is stabilised by a range of mutations and resembles a sparsely-populated state in equilibrium with active HDAC8. Computational models show that the inactive and active states differ by small changes in a regulatory region that extends up to 28 Å from the active site. The regulatory allosteric region identified here in HDAC8 corresponds to regions in other class I HDACs known to bind regulators, thus suggesting a general mechanism. The presented results pave the way for the development of allosteric HDAC inhibitors and regulators to improve the therapy for several disease states.

[1] Institute of Structural and Molecular Biology, Division of Biosciences, University College London, London WC1E 6BT, UK. [2] Department of Chemistry, University College London, London WC1E 6BT, UK. [3] Present address: Nuvisan ICB GmbH, Innovation Campus Berlin, Müllerstraße 178, 13353 Berlin, Germany. [4] These authors contributed equally: Nicolas D. Werbeck, Vaibhav Kumar Shukla. ✉email: d.hansen@ucl.ac.uk

Histone deacetylases (HDACs) have well—established roles in epigenetics and tumour biology, and new cellular roles of these enzymes are being revealed at a remarkable pace. This has led to proposals to target HDACs for the treatment of diseases ranging from cancers to neurodegenerative disorders[1–5]. The zinc-dependent class I HDACs[6] include HDAC1, -2, -3 and HDAC8. HDAC1 and HDAC2 show deacetylase activity, and their activity increases dramatically once bound to inositol phosphates and other HDAC complex subunits, such as MTA1[7], SAP30 and RBBP7[7–10]. While HDAC3 is inactive in isolation, it shows significant activity in complex with the SMRT DAD domain[11]. The isozyme HDAC8 is less active than other class I HDACs, however, HDAC8 displays significant activity on its own and also shows enhanced activity in the presence of metals, such as Fe(II) and Co(II) than Zn(II)[12]. HDAC8 was the first human HDAC for which high-resolution crystal structures were published[13,14]. Consequently, a wealth of structural and computational data is now available for HDAC8, including investigations of the binding of HDAC substrates and inhibitors[15–20] as well as insights into the catalytic mechanism[21].

Despite this, a genuine structure of the free form of HDAC8 (not bound to inhibitor or substrate) has not been reported and differences between active and the inhibited bound states are therefore unknown. Moreover, the mechanism of downregulation by post-translational modification and mutations[7,14,15,22,23] distal to the active site also remain elusive although changes in structural dynamics have been suggested[7,24]. An experimental characterisation of the mechanistic coupling between regulatory perturbations and enzymatic activity is therefore still missing and highly sought after. Addressing this coupling is crucial for our understanding of enzymatic regulation in general, and of our understanding of HDACs in particular in order to fully expand the therapeutic potential of these proteins for the treatment of a plethora of diseases[4,25].

In this work, a bi-directional coupling is revealed between those parts of the HDAC8 enzyme where substrate binding and conversion take place and a distal regulatory region near helix1, loop1 and helix2. Introducing local perturbations in the substrate-binding site by tight-binding inhibitors result in changes near the distal regulatory region whilst mutations introduced in the regulatory region lead to downregulation of HDAC8 activity. An inactive state is stabilised by a range of mutations and this state resembles a sparsely-populated state that is in equilibrium with active, wild-type HDAC8. HDAC8 is therefore allosterically regulated through shifts in populations between exchanging active and inactive states. The active and inactive state of HDAC8 differ by small movements of two α-helices (helix1 and helix2) and a change in the sampling of the loop (loop1) connecting them. As the identified allosteric region coincides with regulatory interactions and regulatory post-translational modifications, these results represent a platform to rationalise how the activity of histone deacetylases can generally be modulated.

## Results

**A distal region is coupled to the active site**. Methyl-TROSY NMR spectroscopy[26] was used to investigate the structural and dynamical coupling throughout the HDAC8 enzyme, Fig. 1. The main focus was on NMR spectra of the δ1-methyl-group of isoleucine side chains because these are highly sensitive reporters of both structure[27] and dynamics[28,29]. The chemical shift assignment was obtained by one-by-one site-directed mutagenesis (Supplementary Figs. 1 and 2) as well as methyl-methyl NOESY experiments[29].

Tight-binding, substrate-competitive HDAC inhibitors were used to provide perturbations near the HDAC8 active site and concomitant changes observed in methyl-TROSY spectra report directly on how these perturbations propagated throughout the enzyme. Three inhibitors with different zinc-binding motif and different modes of binding to the substrate-tunnel were chosen to provide a range of perturbations. Binding of suberoylanilide hydroxamic acid (SAHA) to HDAC8 resulted in small chemical shift changes in the methyl-TROSY NMR spectra, Fig. 1d and Supplementary Fig. 3. The most affected isoleucine residues, I34 and I284, are near the inhibitor-binding site. Binding of the HDAC8-specific inhibitor (R)-2-amino-3-(2,4-dichlorophenyl)-1-(1,3-dihydroisoindol-2-yl)-propan-1-one[30], hereafter named DCPI, also led to small chemical shift changes (Supplementary Fig. 3), with the exception of two residues near the active site, I34 and I269, which both rigidified[27].

In contrast to SAHA and DCPI, binding of the inhibitor Trichostatin A (TSA) with an unsaturated chain and a more polar cap-group, led to substantial and distinct chemical shift changes; Fig. 1c, e. Previous investigations have shown that TSA binds to HDAC8 in a 1:1 stoichiometry in solution[31]. Upon sub-stoichiometric addition of TSA the cross-peaks split into two fractions, revealing that the binding reaction is in the intermediate-to-slow exchange regime[32] in agreement with previous reports[33,34]. Similar peak-splittings, albeit substantially smaller, were observed upon addition of SAHA. Some of the most significant changes upon addition of TSA were, not surprisingly, observed for residues near the active site. A substantial chemical shift change was observed for I284, whilst the cross-peak for I34 disappeared, most-likely due to millisecond dynamics within the HDAC8:TSA complex (see below). More surprisingly, however, chemical shift changes extended to a region remote from the inhibitor-binding interface. This region involves residues in or around helix1 (H1) and helix2 (H2) and comprises I45, I331 and I56, the latter being located 28 Å from the active site (Fig. 1e). Two control experiments were performed to ensure that the observed effects were specific and caused by TSA-binding in the active site. First, TSA was competed out by addition of excess amounts of DCPI, which reversed the TSA-specific effect (Supplementary Fig. 4a). Secondly, a titration of HDAC8 with DMSO did not show any significant change (Supplementary Fig. 4b). Guided by the identified helix1-loop1-helix2 region, additional reporters in this region were incorporated, namely the ε-methyl group of methionine residues M27 and M40. The assignment of these methyl groups was obtained by mutations and both M27 and M40 were severely affected by TSA binding, Supplementary Fig. 5.

**HDAC8 samples sparsely-populated alternative states**. The communication between the active site and the helix1-loop1-helix2 region observed above suggests that the TSA inhibitor stabilises a state of HDAC8, which differs in the helix1-loop1-helix2 region and could potentially already be present in the free form. Multiple-quantum methyl-TROSY CPMG relaxation dispersion experiments[35] are sensitive to μs-ms conformational changes and have the capacity to reveal inter-conversions between states that are only fractionally populated[35,36]. The curved CPMG relaxation dispersions obtained for the free form of HDAC8 reveal a chemical exchange process on the μs-ms time-scale (Fig. 2a and Supplementary Fig. 6) between the observed major state and a low-populated minor state. A global fit of a two-state model to the relaxation dispersions resulted in an overall exchange rate, $k_{ex} = k_{forward} + k_{backward}$, of $1950 \pm 180\,\text{s}^{-1}$ (see "Methods"). In favourable scenarios, the population of the minor state, $p_m$, as well as the site-specific chemical shift differences between the exchanging states, $\Delta\varpi$, can be derived from CPMG experiments[37]. However, because of the large $k_{ex}/\Delta\varpi$ ratios for

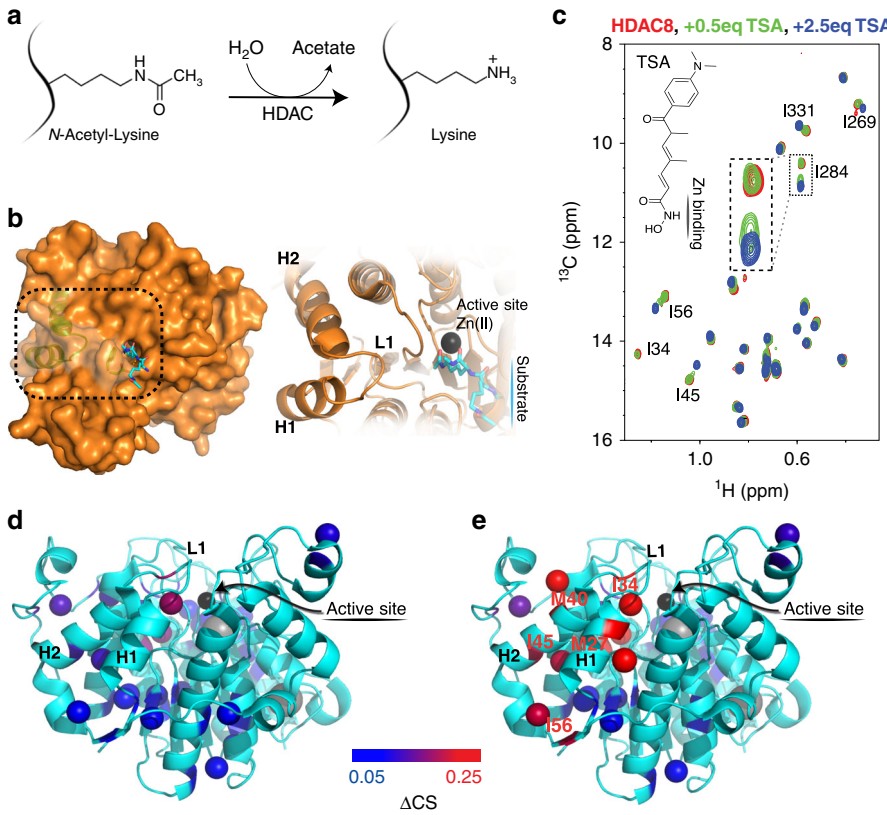

**Fig. 1 Coupling between helix1-loop1-helix2 and the active site. a** The deacetylase reaction catalysed by HDACs. **b** Surface structure of HDAC8 (PDB: 2v5w) with substrate bound (left) and structural elements discussed below highlighted (right). **c** Methyl-TROSY NMR spectra of isoleucine δ1-methyl groups in HDAC8 (30 μM). Overlay of free-form (red), with sub-stoichiometric (14 μM; green) and excess (blue) amounts of TSA (75 μM). Distinct changes are observed for the labelled residues. **d, e** Structural representation of isoleucine and methionine chemical shift changes, ΔCS, upon (**d**) SAHA and (**e**) TSA binding. Long-range effects, up to 28 Å from the binding site, are observed upon TSA binding. The shift changes are calculated as

$$\Delta CS(\text{Ile}) = \sqrt{(\Delta\delta_H/0.28)^2 + (\Delta\delta_C/1.66)^2} \quad \text{and} \quad \Delta CS(\text{Met}) = \sqrt{(\Delta\delta_H/0.38)^2 + (\Delta\delta_C/1.67)^2}$$ based on the standard deviation of assigned chemical shifts[65].

Isoleucine-34 was coloured red in **e**. due to peak disappearance upon saturation with TSA.

free HDAC ($\alpha$-values[38] > 1.2) only the parameter $p_m(1 - p_m)$ $\Delta\varpi^2$ could be obtained accurately. Even so, the sites with the largest $p_m(1 - p_m)\Delta\varpi^2$, including I34 (Supplementary Fig. 6), I56 and M27 (Fig. 2a), are all located around the helix1-loop1-helix2 region, Fig. 2b, and a reasonably strong correlation (Supplementary Fig. 7) is observed between $\sqrt{p_m(1 - p_m)}|\Delta\varpi|$ and the difference in chemical shift between free and TSA-bound HDAC8. Thus, the CPMG relaxation dispersion experiments suggest there is a sparsely-populated conformational state of free HDAC8, which has similar structural characteristics to the TSA-bound form in the helix1-loop1-helix2 region.

Performing CPMG relaxation dispersion experiments on an HDAC8 sample with excess amounts of TSA (1:5 ratio) resulted in large dispersion profiles for many of those sites that were affected by TSA binding (Fig. 2c) and these dispersions report on a unimolecular reaction with an overall rate constant of $k_{ex} = 1100 \pm 50\,\text{s}^{-1}$. In order to exclude the possibility that the observed effect originates from rapid dissociation and association events of inhibitor, rather than a unimolecular structural conversion of the enzyme, the CPMG relaxation dispersion experiments were repeated with increasing concentrations of the TSA inhibitor (Supplementary Fig. 8). If the relaxation dispersions observed for the TSA-bound form were due to a bimolecular binding reaction of TSA one would observe an increase of $k_{ex}$ nearly proportional to the TSA concentration as well as a significant reduction of the population of the minor species, $p_m$ for increased TSA concentrations. When the TSA

concentration was double that of the initial concentration used neither $k_{ex}$ nor $p_m$ changed significantly, (see Supplementary Fig. 8). Thus, the CPMG relaxation dispersions observed for the TSA-bound form of HDAC8 report on a chemical exchange process independent of the TSA concentration and are thus most likely unimolecular in nature.

Methyl-TROSY multi-quantum CPMG relaxation dispersion report predominantly on the difference in the methyl $^{13}$C chemical shift[35], $\Delta\omega_C$, which in turn are good reporters of changes in local structure. Even more pronounced than for the free form, the derived chemical shift differences from the CPMG relaxation dispersion experiments, $|\Delta\varpi_{C,CPMG}|$, correlate well with the differences in chemical shift between TSA-bound and free HDAC8, Fig. 2d. This result strongly suggests that the helix1-loop1-helix2 conformation of TSA-bound HDAC is in exchange with a state that closely resembles the major state in the absence of TSA, that is, the active form.

**Helix1-loop1-helix2 is a regulatory region.** Single-point mutations were introduced in order to characterise states of HDAC8 with the helix1-loop1-helix2 region in a state akin to the TSA-bound form and the sparsely-populated state of free HDAC8. The phosphorylation mimic S39E, which is known to downregulate HDAC8 activity[39,40], the M40A mutant in the vicinity of S39 as well as a double mutant S39EM40A in helix2, were initially considered. In the methyl-TROSY spectra, the cross-peaks of residues in the helix1-loop1-helix2 region that are affected by

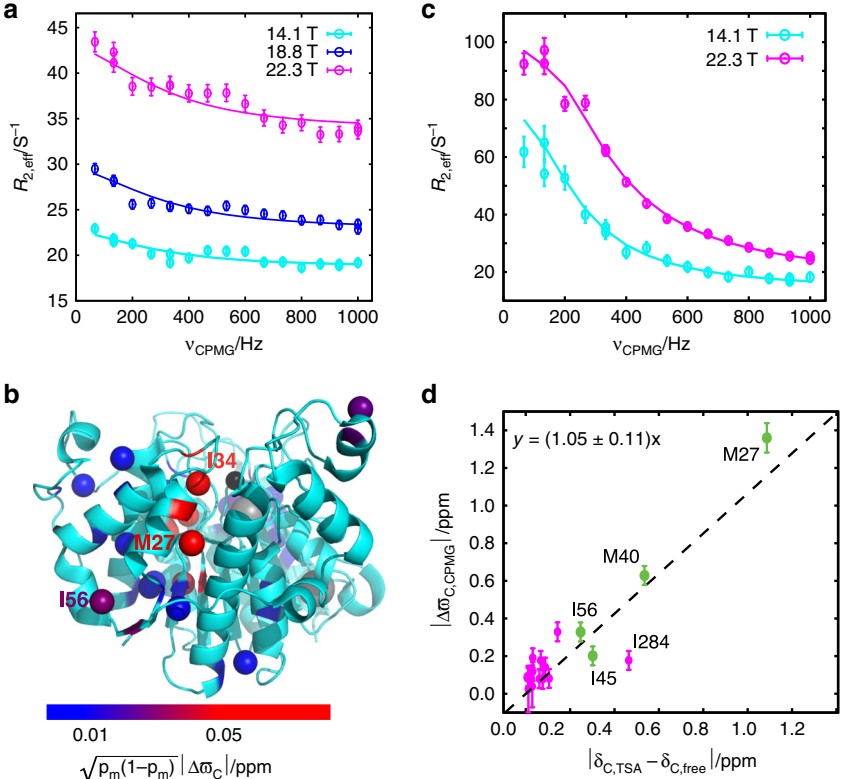

**Fig. 2 HDAC8 is in exchange with an alternative state. a** Methyl multiple-quantum CPMG relaxation dispersion profiles for M27 $^{13}C^{\varepsilon}H_3$ of free HDAC8. The curved relaxation dispersion profiles reveal that the major state is in exchange with a low-populated minor state. **b** Structural representation (PDB: 2v5w[15]) of the $\sqrt{p_m(1-p_m)}|\Delta\varpi_C|$ parameters obtained from the CPMG relaxation dispersions, showing the sites affected by the exchange, Supplementary Table 1. **c** Multiple-quantum relaxation dispersion profiles of M27 $^{13}C^{\varepsilon}H_3$ for TSA-bound HDAC8. In **a**, **c**, circles represent experimental data, vertical lines represent the standard derivation (s.d.) and the solid line is the result of a least-squares fit to a two-state model (see text). **d** Correlation between $^{13}C$ chemical shift differences between free and TSA-bound HDAC8, and $|\Delta\varpi|$ obtained from CPMG relaxation dispersion experiments on TSA-bound HDAC8. Vertical lines represent the standard derivation (s.d.) of the derived $|\Delta\varpi|$ parameters and data points in green are sites located in or near the helix1-loop1-helix2 region.

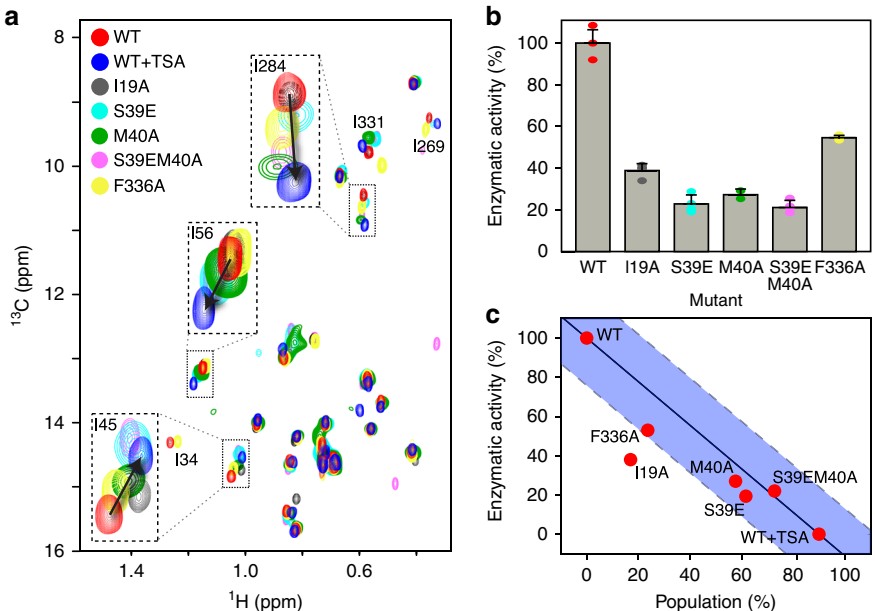

**Fig. 3 The alternative state of the helix1-loop1-helix2 region is inactivating. a** Overlay of methyl-TROSY NMR spectra of WT-HDAC8 (red), WT-HDAC8 with 2.5eq TSA (blue), and the five mutations I19A (brown), S39E (cyan), M40A (green), S39EM40A (magenta) and F336A (yellow). The mutants stabilise the alternate state to varying degrees. **b** Relative enzymatic deacetylase activity of the HDAC8 mutants (see Methods). Vertical lines represent the uncertainty (r.m.s.d.). **c** Relative enzymatic activity *versus* population of the TSA-bound-like state of HDAC8. The full-drawn line connects WT and WT + TSA, while the blue shaded area indicates the r.m.s.d. (±24%).

TSA binding and show relaxation dispersion, i.e. I34, I45, and I56, are also significantly affected by the S39E and M40A mutations, Fig. 3a. Moreover, I269 and I284 near the substrate-binding-site were also shifted and the intensity of I34 and I284 was reduced, which is in agreement with the observations from TSA binding. Activity assays (see "Methods") show that both S39E and M40A have reduced enzymatic activity compared to wild-type, 23 ± 4% and 27 ± 3%, respectively (Fig. 3b).

I19 is located near the N-terminus of helix1, ~15 Å from loop1 and 21 Å from the active site and, in light of the I19S mutation known from Cornelia de Lange syndrome patients[22], the I19A and I19S mutations were introduced. The I19A and I19S mutations had reduced enzymatic activities of 38 ± 3% and 3 ± 1% compared to wild-type HDAC8, respectively. For I19A a change in the chemical shift of I45 was observed as well as significant changes in the intensity of I34 and I284 (Fig. 3a). Expression of I19S was so low that only activity measurements were possible. F336 forms hydrophobic interactions with the helix1-loop1-helix2 region and the F336A mutant showed a reduced activity of 54 ± 2%. Similar to the other mutations, the F336A mutation led to changes in the chemical shifts for residues I34, I45, I269, I284.

The TSA binding, Fig. 1, showed that changes at the active site were propagated through the protein and perturbed residues in the helix1-loop1-helix2 region, whereas the results from the mutations have shown that the converse is also true and that perturbations in the helix1-loop1-helix2 region have a corresponding effect on residues near the active site. Thus, there is a bi-directional, regulatory communication between the helix1-loop1-helix2 region and the active site of HDAC8.

**The alternative state is inactive**. The mutants introduced above led to methyl-TROSY spectra with cross-peak positions between the active wild-type HDAC8 (Fig. 3a, red) and TSA-bound HDAC8 (Fig. 3a, blue). Thus, the mutations stabilise, to varying degrees, states that resemble the TSA-bound form in the helix1-helix2 region and the exchanges between states are in the fast-exchange regime, where peak positions are given by population-weighted averages of the populated states. The population of this TSA-bound-like state in each of the mutants, can be estimated from the chemical shifts observed in the methyl-TROSY spectrum when fast-exchange on the NMR time-scale is assumed[41] (see above). Specifically, for each of the mutants, the population of the TSA-bound-like state was calculated by projecting the peak position of I45, I56, and I284 onto the vector connecting the wild-type and the TSA-bound state, Fig. 3a. Small deviations from exact linear shifts between the wild-type and the TSA-bound state can be ascribed to small changes in chemical shifts due to through-space and non-structural effects from the mutations or the TSA inhibitor. Figure 3c shows that the derived populations correlate well with relative enzymatic activity. This is particularly the case for S39E, M40A, S39EM40A, and F336A, where a linear correlation is observed. Adding excess TSA to the mutants led in all cases to spectra very similar to that of TSA-bound wild-type HDAC, Supplementary Fig. 9, further substantiating that the mutants stabilise to a varying degree an inactive state that bears resemblance to the TSA-bound form. I19A shows a lower activity than expected solely from this model of stabilisation, which indicates that mechanisms beyond a stabilisation of the TSA-bound-like form lead to downregulation of HDAC8 for this mutant.

**Structural transition between active and inactive HDAC8**. Unbiased molecular dynamics (MD) simulations, Fig. 4, were used to characterise the structural transition between active and inactive HDAC8. Based on the methyl-TROSY spectra in Fig. 3,

the S39E mutant was chosen as a representation of a state that is between active (free HDAC8) and inactive HDAC8 (TSA:HDAC8). In all, 4-µs MD simulations of both free WT-HDAC8 and S39E-HDAC8 were performed (see "Methods"), while the crystal structure of TSA-bound HDAC8 (pdb: 1t64) was used as a representation of the TSA:HDAC8 state. The initial 1 µs of these simulations was considered an equilibriation period and was therefore not included in the analysis, Supplementary Fig. 10a, b. Both the WT-HDAC8 and the S39E-HDAC8 simulations deviated from the HDAC8:TSA inactive structure in the helix1-loop1-helix2 region, Supplementary Fig. 10b, but notably less so for the S39E-HDAC8 simulation in agreement with the data in Fig. 3b, c. In the simulations, the transition from the active state (red, WT-HDAC8) to the partly active (cyan, S39E-HDAC8) to the inactive state (blue, HDAC8:TSA) involves movements of helix1 and helix2, Fig. 4a–c. Firstly, there are small movements of the centre-of-masses of both the helices as well as a consistent change in the orientation of helix1 and helix2. The movements in turn lead to a substantial change of the dynamic sampling of loop1, Fig. 4d, e. Whereas loop1 in wild-type HDAC8 samples a very broad range of conformations, Supplementary Fig. 10c, the introduction of the S39E mutation leads to loop1 conformations more distant to both the substrate-binding tunnel and a previously observed substrate-bound structure[15].

## Discussion

Several hypotheses have previously been suggested for the mechanism of the aforementioned phosphorylation-mimicking S39D/E-mediated downregulation of HDAC8[40], including an influence on the release of the acetate product by altering the putative R37 gatekeeper residue[42]. This is despite the fact that crystal structures of inhibitor-bound WT-HDAC8 (PDB: 3RQD) and S39D-HDAC8 (PDB: 4RN2) are 'indistinguishable'[18]. Here, we observe a cooperative effect in methyl-TROSY spectra when an inactive state of HDAC8 is partially stabilised by the S39E mutation as well as other mutations. The small, though significant, chemical shift changes observed strongly suggest that there are only minor differences in the structure between the active and inactive state in helix1 and helix2. Long unbiased molecular dynamics simulations confirm that the structural transition between active and inactive HDAC8 involves a slight movement of helix1 and helix2, but a substantial change in the dynamic sampling of loop1, which has previously been shown to be imperative for substrate binding and activity[6,15].

An examination of regulatory elements in class I HDACs suggests that the helix1-loop1-helix2 region could play a conserved role in mediating regulatory processes. In HDAC8, helix2 bears the phosphorylation site at S39 with phosphorylation[39] and the S39E mutation leading to downregulation. In HDAC1 and HDAC3 this region interacts with deacetylase activation domains (DAD) and regulatory inositol phosphates, respectively[7,9]. This similarity between different class I HDACs suggests that the bi-directional communication between helix1-helix2 and the active site of HDAC8 as well as regulation via the stabilisation of an inactive state with an altered helix1-loop1-helix2 conformation, could well reflect the features of a general mechanism for the regulation of class I HDAC enzymes.

It is becoming increasingly clear that allostery plays an important role for enzyme regulation in general and for the regulation of HDACs in particular. Strategies to characterise the mechanism of allosteric regulation in HDACs have so far been limited, presumably because minimal structural differences have been observed between structures of different inhibitor-bound forms. The insights presented here form the basis for characterising the general regulation of HDACs, thereby suggesting

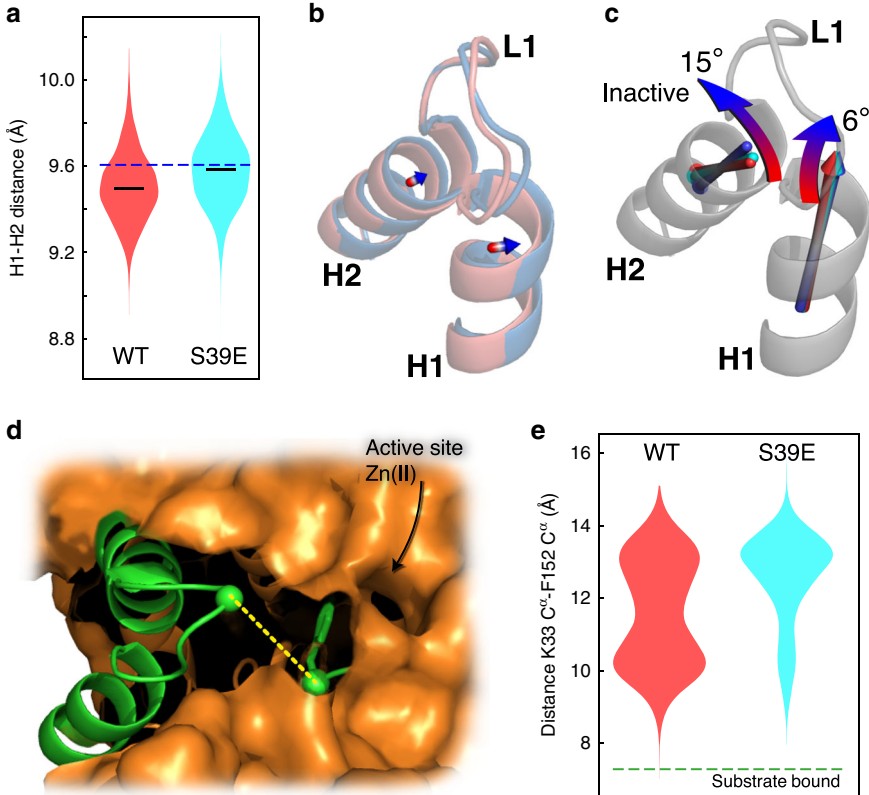

**Fig. 4 Structural changes between active and inactive HDAC8 from molecular dynamic simulations. a** Distribution of the distance between the centre-of-masses of helix1 (H1) and helix2 (H2) for wild-type HDAC8 (left, red) and S39E-HDAC8 (right, cyan). **b** Movement of helix1 and helix2 of active wild-type HDAC8 (red) compared to the crystal structure of TSA-bound HDAC8 (PDB: 1t64; blue). **c** The average orientation of helix1 and helix2 for wild-type HDAC8 (red), S39E-HDAC8 (cyan) and TSA-bound HDAC8. **d** Distance between loop1 and the substrate-binding tunnel, here assessed by the distance (yellow dashed line) between $C^{\alpha}$ of K33 in loop1 and $C^{\alpha}$ of F152, which forms the wall of the substrate-binding tunnel. **e** Distribution of distances between K33 $C^{\alpha}$ and F152 $C^{\alpha}$ for the simulation of wild-type HDAC8 (red) and S39E-HDAC8 (cyan). The distance measured for the crystal structure of substrate-bound HDAC8 (PDB: 2v5w) is shown as a green dashed line.

mechanisms for inhibitor-design, and thus paving the way for new selective HDAC inhibitors to improve the therapy of several diseases.

## Methods

**Protein expression, purification and mutagenesis.** BL21(DE3) strain of *Escherichia coli* cells (New England Biolabs) expressing human HDAC8 were grown at 37 °C in ~99% $D_2O$ minimal M9 media containing $^{15}NH_4Cl$ as the sole nitrogen source and [$^2H/^{12}C$]-glucose as the sole carbon source. Methyl labelling was achieved by addition of 60 mg L$^{-1}$ alpha-ketobutyric acid [U-$^{12}C/^2H$, methyl-$^{13}CH_3$] (for labelling of isoleucines), 90 mg L$^{-1}$ alpha-ketoisovaleric acid [U-$^{12}C/$ $^2H$, methyl-($^{13}CH_3,^{12}CD_3$)] (for labelling of valine and leucine residues) and 150 mg L$^{-1}$ of methionine [$^{13}C/^1H$] (for labelling of methionine residues) 1 h prior to induction. Expression was induced at an $OD_{600nm}$ between 0.5 and 1.0 by 1 mM ITPG. In addition, 0.2 mM $ZnCl_2$ was added before cells were left shaking overnight at 21 °C.

For assignment experiments, the HDAC8 construct described by Vannini et al. with a C-terminal hexa-histidine tag in pET21b expression vector has been used[15]. After cell lysis (lysis buffer: 50 mM Tris–HCl pH 8.0, 3 mM $MgCl_2$, 500 mM KCl, 5 mM imidazole, 5% glycerol, 5 mM beta-mercaptoethanol and 0.25% IGEPAL, small amounts of DNAse, lysozyme and protease inhibitors tablets (1 tablet per 50 ml, Roche)), the protein was purified by Ni-NTA affinity chromatography using a linear imidazole gradient (5–500 mM imidazole in lysis buffer without IGEPAL, DNAse, lysozyme and protease inhibitor). Fractions containing HDAC8 were pooled and dialysed (dialysis buffer: 50 mM Tris–HCl pH 8.0, 150 mM KCl, 1 mM DTT and 5% glycerol) overnight before the sample was buffer exchanged into NMR-buffer (50 mM potassium phosphate, 20 mM KCl, 1 mM DTT and 1 mM $NaN_3$ at pH 8.2) or frozen and stored at −80 °C. Mutations were introduced by the Quikchange approach and verified by DNA sequencing. Primer sequences used for each mutation are given in Supplementary Table 3.

For all other experiments an HDAC8 construct in pET29b+ expression vector has been used[43], which bears a TEV-cleavable N-terminal (His)$_6$-NusA tag. Expression and initial Ni-NTA purification of this construct was similar to that

described above. Following the initial Ni-NTA affinity chromatography, fractions containing HDAC8 were dialysed in cleavage buffer (50 mM Tris pH 8.0, 150 mM KCl, 5 mM β-mercaptoethanol, 5% glycerol). After cleavage with His-tagged TEV protease, the cleaved HDAC8 was separated from the His-NusA-tag, the TEV-protease and nonspecific contaminants by a second Ni-NTA chromatography step. The flow-through was pooled, concentrated and subjected to a gel filtration column (S75, GE-Healthcare) in gel-filtration buffer (50 mM Tris pH 8.0, 150 mM KCl, 1 mM TCEP, 5% glycerol). Finally, the protein was concentrated and buffer exchanged into NMR-buffer.

**NMR experiments.** Unless stated otherwise NMR experiments of HDAC8 were conducted in NMR-buffer (50 mM potassium phosphate, 20 mM KCl, 1 mM DTT and 1 mM $NaN_3$ at pH 8.2) and at 25 °C. Two-dimensional methyl-TROSY spectra were recorded using the experiment described by Tugarinov et al.[26]. Throughout this work, data were recorded on the following spectrometers: Bruker Avance III 500 MHz (room temperature TXI probe), Varian Inova 600 MHz (HCN cryogenic probe), Bruker Avance III 700 MHz (TCI cryogenic probe), Bruker Avance III HD 800 MHz (TCI cryogenic probe) and Bruker Avance III HD 950 MHz (TCI cryogenic probe). Data were processed using NMRPipe[44] and visualised with Sparky[45,46].

Methyl-NOESY experiments ($^{13}C_{methyl}(\omega_1)$–NOESY–$^{13}C_{methyl}(\omega_2)$– $^1H_{methyl}(\omega_3)$ and $^1H(\omega_1)$–NOESY–$^{13}C_{methyl}(\omega_2)$–$^1H_{methyl}(\omega_3)$) for chemical shift assignments were performed using standard pulse sequences at static magnetic fields of 18.8 T and 16.4 T and with a NOESY mixing-time of 200 ms. Methyl-TROSY based MQ-CPMG experiments were performed at different field strengths according to Korzhnev et al. [35] using a constant-time delay for the CPMG period of 30 ms and CPMG frequencies from 20 to 1000 Hz. Peak intensities were obtained using FuDA[47]. The effective transverse relaxation rates, $R_{2,eff}$, were calculated as $R_{2,eff}(\nu) = -\ln(I_\nu/I_0)/T_{relax}$, where $I_\nu$ is the peak intensity at the CPMG frequency $\nu$ and $I_0$ the intensity measured without the delay $T_{relax}$. The rates were used as input for the analysis software CATIA[48] to extract the parameters $k_{ex}$ (exchange rate between minor and major state, a global parameter), $p_m$ (population of minor state, a global parameter) and the chemical shift difference between major and minor state, a residue-specific parameter. A simple multiple-quantum basis-set[35] was used

for integration of the spin-evolution during the CPMG element in CATIA. Errors of the derived parameters were determined using the co-variance method[49].

The obtained multiple-quantum relaxation dispersions depend on both the chemical shift difference for $^{13}$C, $\Delta\omega_C$, and $^1$H, $\Delta\omega_H$, however, since the number of $^{13}$C refocusing pulses is increased during the experiment, while the number of proton refocusing pulses is kept constant, the dependence on $\Delta\omega_H$ is small[35]. F-test statistics[49] was used to determine if the inclusion of $\Delta\omega_H$ in the least-squares fit significantly improved the quality of the fit. In only one case, the relaxation dispersion of I94 for TSA-bound HDAC8, was the inclusion of $\Delta\omega_H$ significant at the 95% confidence level ($\Delta\omega_H = 0.10 \pm 0.01$ ppm) and consequently $\Delta\omega_H$ was kept fixed at 0 ppm for all other residues in the final analysis.

I162 shows large relaxation dispersions in both the free and TSA-bound state, however, we have concluded that these relaxation dispersions originate from another millisecond process, since this residue is insensitive to inhibitor binding and distant to both the allosteric region and active site.

**Inhibitors**. Trichostatin A (TSA) and SAHA were obtained from Sigma-Aldrich; product nos. T8552 and SML0061, respectively. The HDAC8-specific inhibitor (R)-2-amino-3-(2,4-dichlorophenyl)-1-(1,3-dihydroisoindol-2-yl)-propan-1-one (DCPI) was synthesised following a strategy similar to that published by Whitehead et al. [30]. Briefly, to a stirring mixture of isoindoline (0.11 mL, 1.0 mmol), N-Boc-(R)-2,4-dichlorophenylalanine (344 mg, 1.0 mmol), N-ethyl-N'-(3-dimethylaminopropyl)carbodiimide hydrochloride (182 mg, 1.0 mmol) and anhydrous 1-hydroxybenzotriazole (203 mg, 1.5 mmol) in N,N-dimethylformamide (3.0 mL) was added N,N-diisopropylethylamine (1.3 mL, 7.5 mmol). Stirring was continued for 20 h at 20 °C. Evaporation gave a residue that was dissolved in ethyl acetate (20 mL); the organic layer was washed with water (2 × 20 mL) and the combined aqueous fractions were re-extracted with ethyl acetate (4 × 20 mL). The combined organic layers were washed with saturated aqueous sodium hydrogen carbonate (2 × 20 mL), then with brine (20 mL). The resulting organic layer was dried over magnesium sulfate, filtered and evaporated. The residue was purified by column chromatography (5:95 v/v ethyl acetate:dichloromethane) on silica gel to give (R)-tert-butyl (3-(2,4-dichlorophenyl)-1-(2,3-dihydro-1H-inden-2-yl)-1-oxopropan-2-yl)carbamate (330 mg, 76%) as a pale yellow solid. Hydrogen chloride was passed through a stirring solution of (R)-tert-butyl (3-(2,4-dichloro-phenyl)-1-(2,3-dihydro-1H-inden-2-yl)-1-oxopropan-2-yl)carbamate (0.15 g, 0.35 mmol) in diethyl ether (10 mL) for 2 h. The precipitate was filtered under gravity into a flask containing diethyl ether (10 mL) at reflux. The remaining solid was dried under vacuum to give (R)-2-amino-3-(2,4-dichlorophenyl)-1-(isoindolin-2-yl)propan-1-one hydrochloride (114 mg, 87%) as a white solid.

**Activity assay of HDAC8 and its mutants**. Activity assay of the WT-HDAC8 and its different mutants were performed by using Boc-Lys(Ac)-7-amino-4-methyl-coumarin (MAL) as the substrate. MAL assay was performed as described by Kunze et al.[6] with slight modifications. Briefly, for MAL assay the aliquots of wild-type and each mutant of HDAC8 were prepared at 0.2, 0.4, 0.6 and 0.8 µM concentrations in the total reaction volume of 50 µL in assay buffer (50 mM Tris pH 8.0, 137 mM NaCl, 2.7 mM KCl, 1 mM MgCl2, 1 mg mL$^{-1}$ BSA). The stock solution of the MAL substrate was prepared at the concentration of 50 mM in DMSO, which was diluted in assay buffer having enzyme solution to yield a final concentration of 200 µM in the reaction volume. The HDAC8:MAL solutions having different concentrations of the enzyme and similar concentration of MAL were incubated at 25 °C, after which 50 µL of the reaction solution was pipetted on a 96-well white NBS microplate at 0, 10, 20, 30, 40, 50 and 60 min (time interval 10 min), where the wells had already been loaded with 50 µL developer solution (10 mg/mL trypsin and 4 µM DCPI in assay buffer). The fluorescence was measured (excitation = 380 nm and emission = 460 nm) after 30 min incubation at 25 °C on a BMG FLUOstar Optima plate reader. Initial steady-state rates, $v_0$, were obtained for each concentration of the enzyme from the slope of observed fluorescence versus time. Subsequently, $k_{cat}/K_M$ was obtained from the slope of $v_0$ versus enzyme concentration. Errors (r.m.s.d.) in relative enzymatic activities were estimated as the root-mean-square deviation of the activity measured in assays for WT-HDAC8 and each mutant (six time points and four substrate concentrations).

**Molecular dynamics simulations**. To prepare the WT-HDAC8 model the crystal structure PDB: 1T64 was taken[13] and the loops were filled in using MODELLER[50] and the TSA molecule removed from the crystal structure. Following this hydrogen atoms were added and the protonation states were determined using MolProbity[51]. The system was parameterised using the Amber99SB*-ILDN forcefield with TIP3P water[52,53]. To maintain the bound ions (Zn$^{2+}$, K$^+$), harmonic potentials centred around 2.25 Å, with a force constant of 3.960 J mol$^{-1}$ nm$^{-2}$, were placed between the Zn$^{2+}$ cation and H180 N$^{\delta1}$ and between the distal potassium and carbonyl groups of V182 and T179. An in vacuo energy minimisation was initially carried out[54]. Subsequently the system was solvated in a dodecahedral box with a volume of 458.83 nm$^3$. The system was neutralised by adding K$^+$ counter-ions followed by an additional 19 K$^+$ and 19 Cl$^-$ ions. Following this, a second energy minimisation was carried out and two density equilibrations were performed. The first used a Berendsen barostat and thermostat for 100 ps[54]. The second equilibration was carried out with the Parrinello-Rahman barostat[38] and with a Nòse-Hoover

thermostat[55,56] for 100 ps. Finally, a 2 ns NVT step was carried out to equilibrate the systems using the final run conditions with coupling to a velocity-rescaling thermostat[57]. During all steps, the bond lengths were constrained using the LINCS algorithm[58] and electrostatics were managed using the PME algorithm[59]. The length of each time step was 2 fs. The simulations were carried out in Gromacs[60] at a temperature of 300 K and performed on a Dell XPS-tower with 8 Intel i7-7700 CPUs at 3.60 GHz and equipped with a NVIDIA GeForce GTX 1070 graphics card. The S39E-HDAC8 simulation was prepared in a similar fashion after introducing the mutation. Both WT-HDAC8 and S39E-HDAC8 simulations were carried out for 4 µs (ca. 120 days of wall-clock time per simulation), where the first 1 µs was considered as an equilibrium period and therefore not used for analysis.

The trajectories were analysed using the MDAnalysis[61,62] library in python 3.6.9 and using PyMOL[63]. The orientation of helix1 and helix2 was obtained using the *anglebetwenhelices* python module, which is a part of the *psico* PyMOL extension[64].

**Reporting summary**. Further information on research design is available in the Nature Research Reporting Summary linked to this article.

## Data availability

Source data are provided with this paper. Other relevant data are available from the corresponding author upon reasonable request.

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

## Acknowledgements

Prof. Stefania Di Marco is acknowledged for providing the plasmid of the C-terminal hexa-histidine HDAC8 and Prof. Francesco L. Gervasio is acknowledged for helpful discussions. N.D.W. acknowledges the Federation of European Biochemical Societies and Deutsche Forschungsgemeinschaft for postdoctoral fellowships (WE 5563/1-1). M.B.A.K., L.S. and R.B.P. acknowledge the Wellcome Trust for Ph.D. studentships (102404/13/Z/13) and S.G. acknowledges the BBSRC-DTP for a Ph.D. studentship. Access to ultra-high field NMR spectrometers was supported by the Francis Crick Institute through provision of access to the MRC Biomedical NMR Centre and by the University of Oxford Well-come Institutional Strategic Support Fund, the John Fell Fund, as well as the Edward Penley Abraham Cephalosporin Fund, and the Engineering and Physical Sciences Research Council (EP/R029849/1). The Francis Crick Institute receives its core funding from Cancer Research UK (FC001029), the UK Medical Research Council (FC001029), and the Wellcome Trust (FC001029). This research is supported by the Biotechnology and Biological Sciences Research Council (BB/H022570/1 and BB/R000255/1) and the Leverhulme Trust (RPG-2016-268).

## Author contributions

N.D.W., V.K.S., H.Y., R.B.P., S.M. and S.O.R.G. produced isotope labelled HDAC8 and mutants of HDAC8; S.O.R.G. and C.M.M. synthesised the DCPI inhibitor and MAL substrate; N.D.W., V.K.S., M.B.A.K, R.B.P. and S.O.R.G. performed activity assays; N.D.W., V.K.S., M.B.A.K., H.Y., R.B.P., J.K. and D.F.H. performed NMR experiments; V.K.S. and L.S. performed molecular dynamics simulations. N.D.W., V.K.S. and D.F.H designed the research, analysed the data and wrote the paper. All authors discussed the results and commented on the paper.

## Competing interests

The authors declare no competing interests.
