## [Peer Review File · Nature Communications]

REVIEWER COMMENTS

Reviewer #1 (Remarks to the Author):

The manuscript titled "A Distal Regulatory Region of a Class I Human Histone Deacetylase" addresses importance of allosteric effect of two distant α -helices (H1 and H2) on the active site conformation of HDAC8 enzyme.

The authors made an excellent study showing the allosteric property in detailed utilizing all possible experimental and computational methods. Although recent and older work exist in the field, not many scientists brought the hypotheses and data together to make a picture out of accumulative knowledge of the allosteric effect of HDAC Class I enzymes so far.

It is original, nice flow and the title clearly reflects its contents. It is well organized. I believe the manuscript is timely and will get the attention of many researchers studying in this field. The paper is publishable in the present form with the following comments:

Comment:

1. Why is it needed to run 4 micro second simulation? Is the system reaching to equilibrium after such a long time? No explanations are given on this.

2. It would be nicer to see the progress of MD simulations of substrate bound HDAC and TSA bound HDAC enzymes.

For example; Radius of gyration, time versus RMSD graph of equilibrated and crystal structure enzymes and time versus RMSF of substrate bound and TSA bound enzymes especially in the H1 and H2 helices. It would be easier to follow these fluctuations on the graphs over the simulation time.

Reviewer #2 (Remarks to the Author):

In this manuscript, Werbeck et al. describe solution NMR studies exploring intrinsic dynamics of a human histone deacetylase, HDAC8, and show that a lowly populated species of the wild-type protein represents an inactive state of the enzyme – a state that is selected by a common HDAC inhibitor. The results described are of great interest not only to investigators in the field but also to the broader scientific community including enzymologists, NMR spectroscopists, theoretical biophysicists, among others.

The authors use a combination of spin relaxation experiments, mutagenesis, and molecular dynamics simulations to identify an allosteric mechanism of enzyme regulation that has implications for other Class I HDACs because the same region identified in HDAC8 was previously implicated in the positive (allosteric) regulation of enzyme activity by small molecules and associated subunits. However, crystal structures of these HDACs could not explain the mechanism of allostery and it was suspected that the inactive conformation was poorly populated. In a sense, this question could only be answered through studies of HDAC8 as this is the only HDAC that could be expressed in its properly-folded form in E.coli and studied by NMR.

I do have some concerns, but they could be easily addressed through additional experiments or appropriate revisions to the text. Some of these are because the authors are either unfamiliar with the literature or did not describe previous findings accurately.

1. Line 35 ... HDAC1 and HDAC2 show abundant deacetylase activity; it's HDAC3 that is inactive in the absence of the SMRT DAD domain (see Schultz et al., Biochemistry 2004, 43, 34, 11083-

11091)

2. Line 36 ... The activity of HDAC1 and HDAC2 is enhanced several-fold by inositol phosphates and other HDAC complex subunits such as MTA1 (see Millard et al., Mol Cell 2013, 51, 57-67) SAP30 and RBBP7 (see Marcum & Radhakrishnan, JBC 2019, 294, 38, 13928-13938).

3. Line 37 ... HDAC8 is not an "isoform"; it could, however, be called an isozyme.

4. Line 37 ... HDAC8 is the least active HDAC (see Schultz et al., (2004) above) but it shows enhanced activity in the presence of metals such as Fe(II) and Co(II) than Zn(II) (see Castaneda et al., Biochemistry 2017, 56, 42, 5663-5670). But even then, its kcat/KM is orders of magnitude lower than the other Class I HDACs.

5. Line 42 ... Define "active apo form"

6. Lin 84-88 ... The original structure of HDAC8 in complex with TSA featured 2 TSA molecules (PDB ID: 1T64); could the shifting of resonances observed in Fig. 1c during the latter half of the titration be due to binding to a second equivalent of TSA with a lower affinity for the interaction?

7. Line 162 ... How was the enzymatic activity data measured and what does %age activity mean for the various HDAC8 mutants?

8. Line 191-195 ... While this would otherwise be a reasonable approach, given that the peaks even for the annotated isoleucines do not fall along the vector connecting the apo and inhibitor-bound states, it is recommended that independent spin-relaxation measurements be performed for extra validation.

9. Figure 4: From the MD simulations, is it obvious what causes the loop conformation to change its bias towards the inactive conformation; in other words, does the side chain of E39 have a role in this process? If so, what is it?

10. Figure 4a/e: It would be useful to compare the distance distributions from the MD simulations for TSA-loaded WT with those for the apo WT and S39E mutant.

Reviewer #3 (Remarks to the Author):

Werbeck et al present the study on the conformational flexibility of HDAC8. The authors show clearly that HDAC8 is allosterically regulated by shifts in populations between exchanging states. The manuscript is exciting and the study was performed thoroughly. The major result is originated from the fact that chemical shifts acquired from relaxation dispersion of wild type match with chemical shift difference between wild type and TSA-bound forms. The work was carefully conducted and drawn conclusions are appropriate and reasonable. The reviewer finds only a few minor issues as follows:

a. the authors present only chemical shift difference between TSA-bound form and wild type (Figure 2 d). How does the correlation between chemical shifts obtained from relaxation dispersion and chemical shift difference between other compounds and wild type?

b. only carbon chemical shifts have been used for the correlations between chemical shifts obtained from relaxation dispersion and methyl-trosy (Figure 2 d). How does it look with proton chemical shifts?

c. although the reviewer believes that Figure 3c has been obtained based on populations from slopes between chemical shift variances from relaxation dispersion and chemical shifts from

methyl-trosy (Figure S7). But there is no description. The reviewer thinks that the populations obtained from carbon chemical shifts. How is the correlation using proton chemical shifts?

d. Molecular dynamics provide interesting insights for structural changes. However, chemical shifts of methyl groups calculated with the structures from molecular dynamics would be better for direct comparison with experimental chemical shifts.

Reviewer 1:

The authors made an excellent study showing the allosteric property in detailed utilizing all possible experimental and computational methods ... It is original, nice flow ... It is well organized. I believe the manuscript is timely and will get the attention of many researchers studying in this field. The paper is publishable in the present form with the following comments:

1. Why is it needed to run 4 micro second simulation? Is the system reaching to equilibrium after such a long time? No explanations are given on this.

The molecular dynamics simulations shown in the manuscript are, to the best of our knowledge, about 5 times longer than any MD simulation published on a class I HDAC. Our reasons for the long simulations were:

- (i) No genuine crystal structure of free HDAC8 is known. We therefore started the simulations from TSA-bound HDAC8 and equilibrated the system.
- (ii) From RMSD plots we judged that about 1 microsecond was needed for equilibration of the system. Considering this, we aimed to run the simulations for at least 4 microseconds.

We have now included a new figure, *Figure S10a*, to show the RMSD of the full protein and to clarify that the first 1 microsecond is used for equilibration.

In the manuscript a brief comment about equilibration is now included:

“ ... The initial 1 μ s of these simulations was considered an equilibration period and was therefore not included in the analysis, Fig. S10a,b. Both the WT-HDAC8 and the S39E-HDAC8 simulations ...”

2. It would be nicer to see the progress of MD simulations of substrate bound HDAC and TSA bound HDAC enzymes.
 - a. For example; Radius of gyration, time versus RMSD graph of equilibrated and crystal structure enzymes and time versus RMSF of substrate bound and TSA bound enzymes especially in the H1 and H2 helices. It would be easier to follow these fluctuations on the graphs over the simulation time.

Comparisons with an MD simulation of TSA-bound HDAC8 could be informative. However, generating an accurate force field for the TSA inhibitor is not trivial partly because of the large conjugate parts. We therefore chose to use the already available crystal structure of TSA-bound HDAC8 as a reference for this state. We have made this clearer in the revised manuscript:

“... while the crystal structure of TSA-bound HDAC8 (pdb: 1t64) was used as a representation of the TSA:HDAC8 state ...”

We have now included a plot of RMSD to the initial structure *v.s.* simulation time for the wild-type and S39E simulations. The MD simulations are initiated from TSA-bound HDAC8 and the RMSDs therefore report on the RMSD between TSA-bound HDAC8 and the simulations as a

function of simulation time. This figure is included as Fig. S10a in supporting material. We have also included the RMSD calculations specifically for the H1-L1-H2 motif, Fig. S10b.

Moreover, for the other two measurements discussed in Fig 4, that is the H1-H2 distance and the K33 – F152 distance, we have now in Fig. S10 also shown these measurements as a function of simulation time.

*“... **Figure S10** | a. and b. Root mean square deviations (r.m.s.d.) to the initial structure for molecular dynamics simulations of wild-type HDAC8 (red) and S39E HDAC8 (cyan). a. r.m.s.d. calculated for the full structure and b. r.m.s.d. calculated for the H1-L1-H2 motif. Prior to calculation of r.m.s.d. the structures were aligned against the backbone. c. Distance between K33 Ca and F152 Ca for the simulation of wild-type HDAC8 (red) and S39E-HDAC8 (cyan) shown as a function of simulation time. d. Distance between the centre-of-masses of helix1 (H1) and helix2 (H2) for wild-type HDAC8 (red) and S39E-HDAC8 (cyan) shown as a function of simulation time. The grey areas show the part of the simulation considered as equilibration and thus not included in the analysis....“*

Reviewer 2:

The results described are of great interest not only to investigators in the field but also to the broader scientific community including enzymologists, NMR spectroscopists, theoretical biophysicists, among others. ...

I do have some concerns, but they could be easily addressed through additional experiments or appropriate revisions to the text.

1. Line 35 ... HDAC1 and HDAC2 show abundant deacetylase activity; it's HDAC3 that is inactive in the absence of the SMRT DAD domain (see Schultz et al., Biochemistry 2004, 43, 34, 11083-11091).
2. Line 36 ... The activity of HDAC1 and HDAC2 is enhanced several-fold by inositol phosphates and other HDAC complex subunits such as MTA1 (see Millard et al., Mol Cell 2013, 51, 57-67) SAP30 and RBBP7 (see Marcum & Radhakrishnan, JBC 2019, 294, 38, 13928-13938).

We have now included the suggested citations in the revised version of the manuscript:

“... HDAC1 and HDAC2 show deacetylase activity, and their activity increases dramatically once bound to inositol phosphates and other HDAC complex subunits such as MTA1,⁹ SAP30 and RBBP7⁷⁻¹⁰. While HDAC3 is inactive in isolation, and show significant activity in complex form with SMRT DAD domain¹¹. ...”

3. Line 37 ... HDAC8 is not an "isoform"; it could, however, be called an isozyme

We agree with the reviewer. This has been corrected throughout the manuscript.

4. Line 37 ... HDAC8 is the least active HDAC (see Schultz et al., (2004) above) but it shows enhanced activity in the presence of metals such as Fe(II) and Co(II) than Zn(II) (see Castaneda et al., Biochemistry 2017, 56, 42, 5663-5670). But even then, its kcat/KM is orders of magnitude lower than the other Class I HDACs.

We have now added the suggested reference and stipulated that HDAC8 is less active than other class I HDACs as well as the fact that different divalent metal-ions affect the activity. The following has been included in the revised manuscript:

“... The isozyme HDAC8 is less active than other class I HDACs, however HDAC8 displays significant activity on its own and also shows enhanced activity in the presence of metals such as Fe(II) and Co(II) than Zn(II)¹² ...”

5. Line 42 ... Define "active apo form".

We have changed the term ‘active apo form’ to ‘free form’ as used in other places in the manuscript:

“... *free form of HDAC8 (not bound to inhibitor or substrate)*...”

6. Lin 84-88 ... The original structure of HDAC8 in complex with TSA featured 2 TSA molecules (PDB ID: 1T64); could the shifting of resonances observed in Fig. 1c during the latter half of the titration be due to binding to a second equivalent of TSA with a lower affinity for the interaction?

It was shown previously that, in solution, TSA binds HDAC8 in a 1:1 stoichiometric ratio (Singh, Biochemistry 2014). We have now included this reference:

“...*Previous investigations have shown that TSA binds to HDAC8 in a 1:1 stoichiometry in solution*³². ...”

Despite this previous study, we have performed several control experiments to verify that potential low-affinity binding of a second TSA molecule does not interfere with our conclusions.

Firstly, the competition binding in Fig S4a shows that the effect from TSA binding to HDAC8 can be reversed by binding of the DCPI inhibitor. This, in turn, shows that the TSA and DCPI inhibitors bind to the same location in HDAC8 (active site). Had there been unspecific effects in the NMR spectra due to TSA, the DCPI inhibitor would not have been able to reverse these effects, since we believe it is highly unlikely that the very different DCPI inhibitor can bind to the same unspecific site as TSA.

Secondly, the kinetics for the binding and dissociation of TSA is slow, which is in agreement with previously published results, e.g., C. Meyners (2014). For such a case, (slow-exchange regime in NMR), two separate signals are observed in NMR spectra, when both the bound and free species of HDAC8 are present – in agreement with Fig 1c, where separate peaks are observed at 0.5 equivalent. Regarding the potential binding of a second TSA molecule.

- a) If the potential second binding was in the slow-exchange regime (unlikely) it would not affect the analysis.
- b) If the potential low-affinity binding was in the intermediate-exchange regime, line-broadenings and relaxation dispersions would be expected.
- c) If the potential low-affinity second binding was in the fast-exchange regime (the most likely scenario if there was a second binding), peak shifts would be observed.

Re **b**. Intermediate-exchange is ruled out by Fig S8, where we have shown the results of CPMG relaxation dispersions carried out at multiple concentrations of TSA and did not observe any significant change in k_{ex} and p_m .

Re **c**. (i) Fig S4 does not show any shifts between HDAC8+DCPI and (HDAC8+TSA+DCPI), which points to the absence of a second binding (see above). (ii) We have now included an extra panel to Fig. S8, which shows the methyl-TROSY spectrum at three different concentrations of

TSA. This corresponds to $[TSA]_0/[HDAC8]_0$ ratios of ca. 5,10, and 15. No shifts are observed, therefore strongly suggesting that the affinity of the potential second TSA-binding site is so low that it is not affecting the NMR data.

Overall, for the conditions used in our experiments, we have not been able to detect a second binding event of TSA. This strongly points to the fact that the affinity of this potential second binding must be very low and in particular so low that it does not affect our results.

In Figure S8, we have added panel c and the following to the legend:

“ ... c. Methyl-TROSY spectra obtained for samples with HDAC8 and TSA added in ratios of ~1:5 (green), ~1:10 (blue), and ~1:15 (purple). No chemical shift changes are observed, thereby confirming that potential low-affinity binding of a second TSA molecule to HDAC8 does not affect the NMR results.”

In addition to extending Figure S8, we have also transferred an important message about the binding of TSA to HDAC8 into the main text. This is to clearly show in the revised manuscript (main text), that the dynamics observed for the HDAC8:TSA complex is indeed unimolecular in nature. The following has been included.

“ ... In order to exclude the possibility that the observed effect originates from rapid dissociation and association events of inhibitor, rather than a unimolecular structural conversion of the enzyme, the CPMG relaxation dispersion experiments were repeated with increasing concentrations of the TSA inhibitor (Fig. S8). If the relaxation dispersions observed for the TSA-bound form were due to a bimolecular binding reaction of TSA one would observe two effects upon an increase of the TSA concentration, that are, an increase of k_{ex} nearly proportional to the TSA concentration as well as a significant reduction of the population of the minor species, p_m . When the TSA concentration was double that of the initial concentration used neither k_{ex} nor p_m changed significantly, (see Fig. S8). Thus, the CPMG relaxation dispersions observed for the TSA-bound form of HDAC8 are reporting on a chemical exchange process independent of the TSA concentration and are thus most likely unimolecular in nature. ... ”

7. Line 162 ... How was the enzymatic activity data measured and what does %age activity mean for the various HDAC8 mutants?

The sentence that the reviewer is referring to has been made clearer:

“ ... The I19A and I19S mutations had reduced enzymatic activities of $38\pm 9\%$ and $3\pm 1\%$ compared to wild-type HDAC8, respectively ... ”

We have also now provided the details related to activity measurement. %age activity shows the percentage change in k_{cat}/K_M relative to wild-type HDAC8.

“ ... Initial steady-state rates, v_0 , were obtained for each concentration of the enzyme from the slope of observed fluorescence versus time. Subsequently, k_{cat}/K_M was obtained from the slope of v_0 versus enzyme concentration ...”

8. Line 191-195 ... While this would otherwise be a reasonable approach, given that the peaks even for the annotated isoleucines do not fall along the vector connecting the apo and inhibitor-bound states, it is recommended that independent spin-relaxation measurements be performed for extra validation.

It is correct that the chemical shifts do not follow exactly a straight line between the apo state and the inhibitor-bound state. The main reason to this is the fact that the TSA inhibitor itself and each of the mutations can lead to small ‘through space’ changes in chemical shift in addition to the shift corresponding to the structural change.

We have now added a brief explanation of this in the revised manuscript:

“...Small deviations from exact linear shifts between the wild-type and the TSA-bound state can be ascribed to small changes in chemical shifts due to through-space and non-structural effects from the mutations or from the TSA inhibitor. ...”

We agree with the reviewer that having CPMG relaxation dispersion measurements for all the mutants would have been ideal. However, the linear shifts observed in Fig. 3a imply fast exchange between conformations for the mutants, in agreement with a destabilisation of the active state by the mutations and thus a faster rate. This, in turn, means that the exchange falls outside the window, where a quantification using CPMG-relaxation dispersion experiments is possible. Despite this, we did record MQ-CPMG relaxation dispersions for three of the mutants, however, as expected from Fig 3a, the exchange was too fast to be quantified by CPMG.

We still strongly believe that the correlation observed in Fig. 3c strongly support our conclusion. It is worth to note that Fig 3c shows a correlation between two widely different measurements: (i) enzymatic activities obtained from enzymatic assays and (ii) populations obtained from shifts in peak positions in NMR spectra.

9. Figure 4: From the MD simulations, is it obvious what causes the loop conformation to change its bias towards the inactive conformation; in other words, does the side chain of E39 have a role in this process? If so, what is it?

We are currently investigating the exact role of the E39 side-chain for the conformation of the L1 loop and for down-regulation of HDAC8. From the data presented and as stated in the manuscript, the role of the S39E mutation is to stabilise an inactive state of HDAC8. For the exact mechanism, our preliminary data suggests that the side-chain of E39 causes electrostatic repulsion between the two helices, H1 and H2, due to the presence of D29 in H1. However, it is our opinion that more simulations, mutants, and activity assays need to be performed before we confidently can state the exact mechanism of the S39E mutation, apart from stabilising the inactive state relative to the active state of HDAC8.

10. Figure 4a/e: It would be useful to compare the distance distributions from the MD simulations for TSA-loaded WT with those for the apo WT and S39E mutant.

Comparisons with an MD simulation of TSA-bound HDAC8 could be informative. However, generating an accurate force field for the TSA inhibitor is not trivial partly because of the large conjugate parts. Of note is that the simulations that we have performed are, to the best of our knowledge, about 5 times longer than any other simulation on HDAC8 and took *ca.* 120 days of wall-clock time per simulation. We therefore chose to use the already available crystal structure of TSA-bound HDAC8 as a reference for this state and we have added the following to the revised manuscript:

“... while the crystal structure of TSA-bound HDAC8 (pdb: 1t64) was used as a representation of the TSA:HDAC8 state ...”

Reviewer 3:

... The authors show clearly that HDAC8 is allosterically regulated by shifts in populations between exchanging states. The manuscript is exciting and the study was performed thoroughly. The major result is originated from the fact that chemical shifts acquired from relaxation dispersion of wild type match with chemical shift difference between wild type and TSA-bound forms. The work was carefully conducted and drawn conclusions are appropriate and reasonable.

The reviewer finds only a few minor issues as follows:

- a. the authors present only chemical shift difference between TSA-bound form and wild type (Figure 2 d). How does the correlation between chemical shifts obtained from relaxation dispersion and chemical shift difference between other compounds and wild type?

The comparison in Fig 2d was presented in order to show that the TSA-bound state of HDAC8 is in exchange with a state of HDAC8 that resembles the active state in the H1-L1-H2 motif. Moreover, chemical shift differences are observed up to 28 Å from the active site in HDAC8.

On the contrary, binding of the DCPI inhibitor only leads to small changes in chemical shifts, which are all located in the vicinity of the active site (Figure S3b). As such, the DCPI inhibitor does not lead to allosteric structural changes.

Upon binding of the SAHA inhibitor to HDAC8 only small chemical shift changes are observed. The correlation between (x-axis) chemical shift difference between SAHA and wild type and (y-axis) $|\Delta\omega|$ from CPMG relaxation dispersion experiments on HDAC8:TSA for the isoleucine residues of HDAC8 is shown below.

This plot is also in agreement with the manuscript, where we state that the TSA inhibitor is unique in causing large allosteric effects in the H1-L1-H2 motif and the largest chemical shift perturbations. Although we find the above plot interesting, we do not think that it will make our conclusions stronger nor do we think that it will help the general readership of Nature Communications to better follow our paper.

- b. only carbon chemical shifts have been used for the correlations between chemical shifts obtained from relaxation dispersion and methyl-trosy (Figure 2 d). How does it look with proton chemical shifts?**

Although MQ-TROSY CPMG relaxation dispersion experiments, theoretically, also report on the proton chemical shift differences, the dependences on these are very small (see e.g. Korzhnev, J. Am. Chem. Soc., 2004). Still, we did attempt to extract the proton $\Delta\omega$ from the MQ-TROSY CPMG relaxation dispersions, but accurate values could not be obtained - in agreement with the original findings by Korzhnev et al. We did also try to obtain the proton $\Delta\omega$ from triple-quantum relaxation dispersions (Yuwen et al. Angw. Chem., 2016), however, the spectra did not provide enough signal-to-noise to allow for these to be determined.

More importantly is that ^{13}C chemical shift, in particular aliphatic ^{13}C chemical shifts, report accurately on local structure, whereas ^1H chemical shifts depend substantially more on long-range interactions such as ring-current effects and electrostatics. Using ^{13}C chemical shifts is therefore accurate for probing small changes in structure.

We have now added the following to the revised manuscript to make this clear.

“ ... Methyl-TROSY multi-quantum CPMG relaxation dispersion report predominantly on the difference in the methyl ^{13}C chemical shift³⁵, $\Delta\omega_{\text{C}}$, which in turn are good reporters of changes in local structure. ... ”

- c. although the reviewer believes that Figure 3c has been obtained based on populations from slopes between chemical shift variances from relaxation dispersion and chemical shifts from methyl-trosy (Figure S7). But there is no description. The reviewer thinks that the populations obtained from carbon chemical shifts. How is the correlation using proton chemical shifts?**

The fact that peaks move on a line between free (enzymatically active) HDAC8 and inactive TSA-bound HDAC8 (Fig 3a) strongly indicate that the active and inactive states of HDAC8 are populated to varying amount and also that the exchange between states is in the so-called fast-exchange regime.

This has now been further stressed in the revised version:

“... region and the exchanges between states are in the fast-exchange regime, where peak positions are given by population-weighted averages of the populated states...”

The fast-exchange regime implies that the position of the peaks observed in the Methyl-TROSY spectra is a population-weighted average of the chemical shifts in the populated states. In turn, this means that the population of the active state of HDAC8 can be obtained from the position (^{13}C and ^1H) of the peaks. How the populations were derived is explained in the revised manuscript:

“ ... for each of the mutants, the population of the TSA-bound-like state was calculated by projecting the peak position of I45, I56, and I284 onto the vector connecting the wild-type and the TSA-bound state, Fig. 3a. ...”

Therefore, both the ^{13}C and the ^1H chemical shift changes are used to derive the populations. It is important to note, however, that most of the chemical shift changes are dominated by a shift in ^{13}C , which in turn report on (small) changes in local structure.

d. Molecular dynamics provide interesting insights for structural changes. However, chemical shifts of methyl groups calculated with the structures from molecular dynamics would be better for direct comparison with experimental chemical shifts.

We agree that a direct comparison between chemical shifts calculated from three-dimensional structure(s) and the shifts observed with NMR would be ideal. However, generally, the uncertainty in predicting methyl chemical shifts from a three-dimensional structure is substantially larger than the chemical shift changes that we observed, *ca.* 1.1 ppm for ^{13}C (Hansen et al. J. Am. Chem. Soc. 2010) and *ca.* 0.17 ppm for ^1H (Sahakyan, J. Biomol NMR, 2011). We did still try to predict the chemical shift changes between wild-type and S39E HDAC8 from the molecular dynamics simulations, but unfortunately – as expected – the uncertainties were substantially larger than the shifts observed.

REVIEWERS' COMMENTS:

Reviewer #2 (Remarks to the Author):

In the revised manuscript, the authors have addressed all my concerns; the manuscript is suitable for publication.